Systematic review of electron transfer study in DNA relevant to Parkinson’s disease and scanning tunneling microscopy

Che Lah Muhammad Hanif 1 2 3
Faruque Reza Mohammed 1 3
Shamsuddin Shaharum 3 4 5
Watanabe Isao 1 2 5 6
Abdullah Jafri M. brainsciences@gmail.com 1 3
1 Department of Neurosciences, School of Medical Sciences, Universiti Sains Malaysia , Kubang Kerian , Kelantan , Malaysia
2 Meson Science Laboratory, RIKEN Nishina Center, Institute of Physical and Chemical Research (RIKEN) , Wako , Saitama , Japan
3 Brain and Behaviour Cluster, School of Medical Sciences, Universiti Sains Malaysia , Kubang Kerian , Kelantan , Malaysia
4 School of Health Sciences, Universiti Sains Malaysia , Kota Bharu , Kelantan , Malaysia
5 USM-RIKEN Interdisciplinary Collaboration for Advanced Sciences, Universiti Sains Malaysia , Gelugor , Penang , Malaysia
6 School of Distance Education, Universiti Sains Malaysia , Gelugor , Penang , Malaysia
Uversky Vladimir
Electronic publication date: 2025 Aug 20
Publication date: 2025
Volume: 13
Electronic Location ID: e19807
Received 2024 Jun 17; Accepted 2025 Jul 7
Copyright: ©2025 Che Lah et al.
Copyright year: 2025
Copyright holder: Che Lah et al.
License: This is an open access article distributed under the terms of the Creative Commons Attribution License, which permits unrestricted use, distribution, reproduction and adaptation in any medium and for any purpose provided that it is properly attributed. For attribution, the original author(s), title, publication source (PeerJ) and either DOI or URL of the article must be cited.
License URL: https://creativecommons.org/licenses/by/4.0/

Keywords: Parkinson’s disease, Electron transfer, DNA, Scanning tunneling microscopy

Funding: The Ministry of Higher Education of Malaysia Skim Latihan Akademik Bumimputera KPT(BS)860326295217 Academic Staff Training Scheme KK/PTJPK/JPEND/PSM211 USM(KS)4148 The International Associate Program (IPA) RIKEN Muhammad Hanif Che Lah is supported by scholarships from the Ministry of Higher Education of Malaysia Skim Latihan Akademik Bumimputera KPT(BS)860326295217 and Academic Staff Training Scheme KK/PTJPK/JPEND/PSM211 USM(KS)4148, and was previously supported by the International Associate Program (IPA) RIKEN. The funders had no role in study design, data collection and analysis, decision to publish, or preparation of the manuscript.

==============================
Background

Parkinson’s disease (PD) is the most typical neurological disorder associated with aging in humans. Since PD has much to do with the medical field, most research studies focus on the biological, chemical, and medical aspects of the investigations, in addition to epidemiological studies, drug intervention studies, and much more. The lack of studies using scanning tunneling microscopy (STM) to investigate the electron transfer properties of DNA in PD opens up a new opportunity to look at electron transfer, which is fundamental to understanding the biological processes of the damage-repair mechanism of DNA in this disease, from a physical perspective. Hence, this systematic review was conducted to identify the methods or techniques currently used in the medical-related fields to study electron transfer in PD. related to electron transfer and PD.

Methodology

Scopus, ScienceDirect, and EBSCOhost MEDLINE databases were used to search for literature related to electron transfer and PD.

Results

From the thirty studies identified, PD appears to be caused by various causes, including increased levels of cytochrome c, reactive oxygen species produced by the mitochondria, dysfunction of complex I that interferes with the electron transfer process, and mitochondrial dysfunction triggered by PINK1 mutation. 6.7% of prior research has focused on utilizing DNA as a specific sample for investigating electron transfer in synthetic DNA through the use of STM. This highlights a notable lack of research into the potential of DNA in PD, despite the theoretical advantages that STM offers.

Conclusions

We propose using STM as a new technique to study electron transfer in the DNA of PD from the physics perspective.

Introduction

Parkinson’s disease (PD) is the most widespread neurodegenerative motor disorder associated with aging in humans. PD has the fastest rising prevalence of any neurological ailment included in the Global Burden of Disease, Injuries, and Risk Factors Study, 61% rise in crude prevalence from 1990 to 2016, with population aging playing a significant role (GBD 2016 Parkinson’s Disease Collaborators, 2018). By the 2030s, Asia will have more than 60% of the world’s population aged at least 65 years, with China alone expected to have roughly five million PD patients by 2030, accounting for 60% of all PD patients (Lim et al., 2019).

PD is a gradual irreversible neurodegenerative disorder that affects adults usually over the age of 60, causing the patient to have clinical symptoms such as abnormalities of movement, including slow movements, difficulties with gait and balance, and tremor (Sveinbjornsdottir, 2016). PD is mostly prevalent in older adults at the ages of 85–89 and this abnormally aging condition of the brain might be due to several environmental factors such as pesticides, herbicides, and heavy metals from industrial waste (Caudle et al., 2012; Sveinbjornsdottir, 2016).

PD can also occur in younger adults, at the earliest at the age of 21 (Quinn, Critchley & Marsden, 1987), which is primarily due to the genetic factor of abnormal or damaged deoxyribonucleic acid (DNA) passed on from parents to their children. Many DNA studies on PD are based on the detection of DNA amplifications using the molecular cloning technique, but only at the time when the disease is already showing symptoms due to the degeneration of more than 50 to 80% of the dopaminergic neurons (Paisán-Ruíz et al., 2004; Ay et al., 2015; DeMaagd & Philip, 2015). In PD, where neurodegeneration is observed, the preservation of genomic integrity depends on DNA repair mechanisms, which in turn depend on electron transfer, a process that plays a crucial role in various biochemical functions.

The electron transport process is of fundamental importance for understanding the biological processes of the damage repair mechanism in DNA. Electron transport within DNA involves the movement of electrons through the DNA molecule, which can occur via various mechanisms, including hopping between nucleobases and through π-stacking interactions. This process is essential for the recognition and repair of damaged DNA bases, as it allows repair proteins to detect lesions and coordinate repair activities across the genome (Genereux, Boal & Barton, 2010; Ito et al., 2012). For instance, the ability of DNA to transport charge can facilitate the signaling between repair enzymes, enabling them to locate and respond to sites of damage effectively (Sontz et al., 2012; Bartels et al., 2017). While mitochondrial electron transport is vital for cellular energy production and overall cellular health, the electron transport within DNA serves a different purpose, focusing on the maintenance and repair of the genetic material itself. The two processes are interconnected in the broader context of cellular function, as mitochondrial dysfunction can lead to increased oxidative stress, further exacerbating DNA damage (Gorman et al., 2012; Taylor et al., 2024). However, the electron transport process discussed in the context of DNA repair specifically refers to the intrinsic properties of DNA and its ability to facilitate charge transfer.

Although a number of experimental studies have been conducted on electron transfer in DNA (Giese, 2002; Zheng et al., 2023), still little is known about this fundamental matter in which the mechanism of abnormal or damaged DNA is related to PD. As PD has a lot to do with medicine, most research studies tend to focus on the biological and medical aspects of studies, apart from epidemiological studies, drug intervention studies, and much more. Further research is needed to understand how DNA can conduct electricity and facilitate electron transfer and what applications are possible in developing DNA-based therapies for PD.

Scanning tunneling microscopy (STM) is a method that allows direct observation of individual biomolecules at the molecular level, and STM has greatly influenced physics and surface science (Rodríguez-Galván & Contreras-Torres, 2022). To our knowledge, the potential of STM to study the electron transfer properties of DNA in PD has not yet been recognized. This opens up a new opportunity to look at electron transfer, which is fundamental to understanding the biological processes of the DNA damage repair mechanism in this disease, from a physical perspective. To fill the above-mentioned gap, the current systematic review study analyzed research in various fields that has been published on PD. The aim was to discuss the role of electron transfer in the factors leading to PD and how electron transfer can be studied in the context of the electronic properties of PD DNA using STM.

Methodology

The current review was performed following the Preferred Reporting Items for Systematic Reviews and Meta-Analyses (PRISMA) guidelines (Moher et al., 2009) including the PRISMA 2020 Checklist (Page et al., 2021; File S1). A search technique was devised utilizing the Scopus, ScienceDirect, and EBSCOhost MEDLINE databases, without imposing any set time frame, to locate all pertinent studies concluded until August 2023.

Search strategy

For this systematic search, we developed a search strategy to identify relevant literature. The search strategy utilized three databases, which were Scopus, ScienceDirect, and EBSCOhost MEDLINE by creating general variable search syntax with the following keywords in each database: “electron transfer”, “DNA”, “genetic”, “scanning tunneling”, and “Parkinson’s disease”. The selection of these databases is justified by their relevance, the quality of the content and the focus on the interdisciplinary nature of the research topics in chemistry, biology, and medicine. Together, these databases provide a sufficient general basis for exploring the complex relationships between electron transfer in DNA, oxidative stress, and neurodegenerative diseases. Although the initial outcomes would be limited, the strategic choice of these databases is consistent with the goals of the systematic literature review. However, future research efforts could benefit from a broader selection of databases to ensure a more comprehensive exploration of the literature.

In Scopus (https://www.scopus.com/search/form.uri#basic), we inserted the following several search terms to address the scope of the systematic review paper: TITLE-ABS-KEY (“electron transfer” “Parkinson’s disease”), TITLE-ABS-KEY (“electron transfer” “scanning tunneling” DNA) and TITLE-ABS-KEY (genetic “electron transfer” “Parkinson’s disease”). The search results were 109, 22, and 12 documents, respectively. The total 143 documents were then checked for duplicates, reprints, book chapters, and books, in which a total of 20 documents were considered for exclusion (Fig. 1). ScienceDirect (https://www.sciencedirect.com/) database was used as the second database to search for the literature for the same research topic. The search terms were: (DNA with Title, abstract, keywords: “electron transfer” “Parkinson’s disease”). From the search, 13 documents were obtained and from that, books chapters and short communication (n = 3) were totally excluded. For the third database, EBSCOhost MEDLINE (https://www.ebsco.com/) the search terms used were: (“electron transfer” “Parkinson’s disease” DNA). The result was eight documents, all from academic journals, without any exclusion.

Figure 1 PRISMA flowchart of the article selection process.

Adapted from Moher et al. (2009).

Combining three databases search results, a total of 141 documents were obtained. In the early screening step, duplicate records from the three databases were checked and 18 documents were excluded, leaving altogether 123 documents for further exclusions in eligibility and inclusion steps (Fig. 1). Two authors (MHCL, IW) independently assessed the studies for eligibility and selected the full texts based on the inclusion and exclusion criteria. Any doubts or disagreements in the selection of articles were resolved by discussion and arbitration by another author (MFR).

Selection criteria

Article inclusion and exclusion criteria were based on PRISMA (Liberati et al., 2009; Moher et al., 2009). The search focused on mapping the existing literature on the electron transfer study in the field of genetic PD. The search then narrowed to the subject areas of exclusively in DNA. The search span was from the year 1991 to 2021. All articles before 1991 were excluded from the search. Articles were included in the systematic review if they met the following criteria: (a) The article was written in English. (b) The original research article was published in a peer-reviewed journal. (c) The article reported on a study of electron transfer in PD. (d) The article specified the genetic relationship to PD and STM.

Quality assessment

The study is based on original research and review articles. All duplicate studies were thoroughly screened to ensure the quality of the systematic review. The abstracts of the articles were thoroughly scrutinized for genetic or DNA-related issues associated to PD. The following exclusion criterion was to limit the papers published in English. In addition, after filtering out duplicate records, other types of articles were removed from the study, with the exception of original research journals and reviews. We selected the articles after evaluating each article based on the inclusion and exclusion criteria mentioned above. The process of literature selection and exclusion at each stage, as described in the PRISMA statement, is shown in Fig. 1.

Data extraction

In the data extraction phase, 30 articles were selected and the following information was extracted: (a) Study design of the original research and review papers. (b) Techniques used in the research. (c) Samples or models used in the research. (d) Findings of the research.

Results

As shown in Fig. 1, we obtained 141 articles from the three database searches in the identification phase, and after removing duplicates, 123 articles remained (File S2). To determine which articles were suitable for this study, we reviewed the titles and abstracts to determine whether the findings met our inclusion and exclusion criteria. After this evaluation, a total of 30 articles remained, which are listed in Table S1 of File S3.

Figure 2A shows the distribution of techniques related to electron transfer used by researchers to study PD. Most of the techniques used are based on biochemical tests and histochemistry, each accounting for 20%. The next two most commonly used techniques are spectrophotometric assay and electrochemistry, accounting for 16% and 12% respectively. Electron transfer assay and electron microscopy contribute 8% and 6% respectively to the distribution of techniques used to study PD related to electron transfer. STM, mass spectroscopy, and magnetic resonance imaging each account for 4% of the techniques used. The least used techniques to study PD are atomic force microscopy, autoradiographic examination, and wing phenotype examination, which all account for 2% of the distribution.

Samples of human origin, which accounted for 47.4% of the samples used in the studies, are the most frequently used samples in the selected studies of PD (Fig. 2B). The second largest proportion, accounting for 42.1% of the samples used, came from animal samples. The least used samples in the selected studies were synthetic samples, which accounted for 10.5%. The brain is the most commonly used sample type in humans and animals, accounting for 64% of all samples used in PD studies (Fig. 2C). The brain is an important organ for PD research, as the disorder mainly originates there and affects the neurons that produce dopamine in a specific region of the brain, the substantia nigra. Depletion of these neurons leads to a decrease in dopamine levels in the brain, which in turn leads to the typical symptoms of the disease such as stiffness, tremors, and reduced mobility. In PD studies, the heart is the second most common sample type at 12%. The three sample types liver, muscle, and blood each account for 8% of human and animal samples used in PD electron transfer studies.

Figure 2D shows that mitochondria, which make up for 26.7% of the total target sample data, are the most commonly used target sample for PD studies. Cell cultures and α-synuclein are the second most frequently used target samples, each accounting for 13.3%. Of these PD-related studies, dopamine is the third most frequently used target sample at 10%. DNA is considered an optimal model for studying electron transport in molecule-conductor systems due to its stability (Xie et al., 2021), readily regulated structure (Artacho et al., 2003), and anticipated electrical characteristics (Porath et al., 2000). However, in this review, only a tiny fraction of 6.7% of previous research using DNA as a target sample for studying electron transfer in synthetic DNA was performed using STM, indicating a significant gap in exploring the potential of DNA in PD despite the theoretical advantages of STM.

Figure 2 Overview of experimental techniques and sample characteristics in the electron transfer and PD studies.

(A) Distribution of different experimental techniques. (B) Proportions of synthetic, human-originated, and animal-originated samples. (C) Types of human and animal tissue samples. (D) Fractions of biological target samples.

Discussion

The main objective of the present systematic review was to select and analyze the various papers on electron transfer in DNA, PD, and STM. For this purpose, 30 eligible papers were screened and analyzed. This section is dedicated to the discussion of the main results of this review.

PD is a highly heterogeneous disorder caused by a mixture of hereditary and environmental factors, and there is no clear etiology for PD. Numerous factors are thought to be clinically important in the pathophysiology of PD. Elevated cytochrome c levels, reactive oxygen species produced by mitochondria, complex I dysfunction leading to disruption of the electron transfer mechanism in defective respiratory complex I, and PINK1-induced mitochondrial dysfunction are among the major factors contributing to PD. We discuss how each factor interacts closely with the others.

Reactive oxygen species generated by mitochondria

Mitochondrial dysfunction is recognized as a common cellular basis for countless human diseases, including classical mitochondrial encephalomyopathies, diabetes, deafness, neurodegenerative diseases, and cancer (El-Khoury et al., 2014; Camargo et al., 2018). The mitochondria are the powerhouse of every cell in our body, generating energy through cellular respiration. However, this process also produces reactive oxygen species (ROS), which can damage various parts of the cells, including proteins, lipids, and DNA. Dysfunctional mitochondria can lead to reduced energy production and cell function.

The decreased electron transfer rates in complexes I and IV, together with other mitochondrial activities, are considered to be the main cause of impaired brain mitochondrial function during aging (Navarro & Boveris, 2010; Jiang et al., 2016). An imbalance between ROS production and antioxidant defense leads to oxidative stress, which promotes various pathophysiological conditions such as diabetes, cancer, arthrosclerosis, aging, Alzheimer’s, and Parkinson’s disease (Umeno, Biju & Yoshida, 2017). These oxidative reactions in cells can produce superoxide anions and hydrogen peroxide (H2O2) in large quantities. A reduced metal ion, such as Fe2+ or Cu+, can react with H2O2 and produce a highly reactive hydroxyl radical (Fato et al., 2008). While most products of oxidative metabolism are beneficial, an imbalance in ROS homeostasis leads to oxidative stress and triggers disease. In PD, these findings suggest that antioxidant defenses are overwhelmed, leading to increased oxidative stress.

Electron leakage upstream of the rotenone binding site in complex I (Hensley et al., 1998) and impaired distal electron transfer in complex I significantly increase H2O2 production. Therefore, mild genetic or acquired complex I abnormalities that are inadequate to affect respiration may result in persistent upregulation of ROS generation (Greenamyre et al., 2001). Neurons, as postmitotic cells that are routinely exposed to high oxygen tensions and calcium levels, may be more vulnerable to the accumulated oxidative stress during their lifetime.

Increased amount of the heme protein cytochrome c in α-synuclein aggregation

Heme is a precursor to hemoglobin, which is an essential protein in red blood cells. Hemoglobin is responsible for the binding of oxygen in the lungs and its transportation to the rest of the body via the bloodstream. The basic structure of the heme consists of Fe2+ that can be found in animal-derived foods. Iron in the heme is an intrinsic generator of ROS. Elevated iron levels enhance neurotoxicity through the generation of hydroxyl radicals, leading to glutathione depletion, protein aggregation, lipid peroxidation, and nucleic acid alterations (Núñez et al., 2012).

A heme iron center is essential for the function of cytochrome c, which has two clearly defined physiological functions: regulating electron transfer in mitochondria and mediating apoptotic cell death. Most of cytochrome c in mitochondria is tightly packed in the inner membrane along with other components of the electron transport chain (Jiang et al., 2016). Since cytochrome c plays a crucial role in electron transfer between the III (ubiquinol:cytochrome oxidoreductase) and the IV (cytochrome oxidase) complexes, dysfunction of cytochrome c molecules can lead to the production of ROS in mitochondria, thereby worsening intracellular conditions for oxidative stress (Shigenaga, Hagen & Ames, 1994). When both cytochrome c and H2O2 were present, α-synuclein clumped together more easily, forming dimers and insoluble clumps.

α-Synuclein is a presynaptic neuronal protein that is genetically and neuropathologically associated with PD. Abnormal soluble oligomeric conformations of α-synuclein known as protofibrils are the toxic species that cause disruption of cellular homeostasis and neuronal death by affecting a variety of intracellular targets, including synaptic function (Stefanis, 2012). The synaptic effects of α-synuclein overexpression include loss of presynaptic proteins, reduced neurotransmitter release, expansion of synaptic vesicles, and suppression of synaptic vesicle recycling. Such deficits are likely to precede overt neuropathology and contribute to synaptic and neuritic degeneration, but the exact point in the cascade at which α-synuclein assumes its neurotoxic potential remains elusive.

Complex I malfunction in Parkinson’s disease

NADH:ubiquinone oxidoreductase, known as complex I, is located in the inner mitochondrial membrane and extends into the mitochondrial matrix (Greenamyre et al., 2001). It is an important component of energy metabolism, as this is where most of the electrons enter the respiratory chain and it is the largest and most complex element of the respiratory chain.

Abnormalities of mitochondrial complex I have been implicated in severe, infantile, and childhood neurological disorders as well as late-onset neurodegenerative diseases, such as PD. The exact role of complex I in these illnesses remains controversial, and the mechanisms by which complex I defects cause neurodegeneration are not entirely clear (Greenamyre et al., 2001). Since complex I is a critical component of the electron transfer chain that contributes to adenosine triphosphate (ATP) synthesis and mitochondrial maintenance, severe defects in complex I activity can lead to reduced ATP synthesis, graded mitochondrial depolarization, and calcium dysregulation. Mild genetic or acquired defects of complex I can also lead to chronic up-regulation of ROS production and may contribute to chronic oxidative insult in postmitotic cells like neurons. These cumulative effects may predispose neurons to calcium-mediated damage and result in neurological diseases.

Epidemiological and biochemical studies of post-mortem brain specimens and peripheral tissues from PD patients have revealed that minor systemic complex I abnormalities have a role in the etiology of the disease (Greenamyre et al., 2001). Since 2006, the standardized incidence rates of PD are eight to 18 per 100,000 person-years, and prospective population-based studies indicate that the occurrence of PD is infrequent prior to the age of 50, but there is a significant rise in prevalence after the age of 60 (De Lau & Breteler, 2006). Epidemiological studies indicate that 10% of PD cases are strictly familial, while the majority are sporadic (Thomas & Beal, 2007).

PINK1-induced mitochondrial dysfunction

PINK1 is a protein encoded by the PINK1 gene, associated with the genetic causes of PD. The PINK1 gene is located on chromosome 1p35-p36, which is part of the nuclear DNA. Mutations in this gene can cause early-onset PD, specifically an autosomal recessive form of the disorder (Rogaeva et al., 2004). While PINK1 is encoded by nuclear DNA, its primary function and the consequences of its mutations are manifested in the mitochondria. The mitochondrial dysfunction observed in PINK1-related PD is not due to mutations in mitochondrial DNA itself but rather stems from the failure of the mitochondrial quality control mechanisms that PINK1 regulates (Abramov et al., 2011; Chin et al., 2023). This distinction is crucial; it indicates that while the genetic basis of PINK1 mutations originates from nuclear DNA, the pathological effects are predominantly mitochondrial.

As a key regulator of mitochondrial health, the importance of PINK1 lies in its involvement in the identification and removal of damaged mitochondria through autophagic mechanisms (Li et al., 2023). Mutations or deficiencies in PINK1 have been linked to impaired mitophagy, resulting in the accumulation of dysfunctional mitochondria and subsequent neurodegeneration observed in PD. Temelie, Savu & Moisoi (2018) proposed a model that PINK1 and Parkin constitute the main system for sensing and modulating removal of dysfunctional mitochondria. Under basal conditions in healthy mitochondria, PINK1 is completely imported into the mitochondrial matrix, where it is immediately destroyed by proteolysis (Yamano & Youle, 2013; Temelie, Savu & Moisoi, 2018). Recessive mutations of the PINK1 gene are the second most common factor in causing autosomal recessive early-onset PD, which is an incurable neurodegenerative movement disorder (Lill, 2016). PINK1-Parkin signaling regulates axonal mitochondrial redistribution in human dopaminergic neurons in response to reduced mitochondrial membrane potential, which is impaired in patients with PINK1 and Parkin mutations (Imai, 2020). However, a previously uncharacterized function for the mitochondrial inner membrane lipid cardiolipin increment has been found to rescue PINK1-induced mitochondrial dysfunction (Vos et al., 2017).

Scanning tunneling microscopy in DNA of Parkinson’s disease

STM is a useful technique for studying the detailed properties of material surfaces at the nanoscale. This technique is based on the phenomenon of quantum tunneling in the field of quantum mechanics. In tunneling, electrons are emitted from a sample when a thin sharp probe is brought very close to the sample surface. The tunneling current is generated when the electrons cross a potential barrier and reach the tip of the probe. To ensure a uniform tunneling current, the distance between the tip and the sample surface is adjusted so that the probe maintains this distance precisely while scanning the surface synchronously. This scanning allows the STM to achieve atomic resolution, although it is mainly suitable for conductive materials. Rodríguez-Galván & Contreras-Torres (2022) demonstrate that the STM can be used to observe biological molecules, particularly in the study of double-stranded DNA, which was the first biomolecule to be studied with the STM. This ability to provide detailed spatial information about surface-bound biomolecules is particularly advantageous when studying the intricate structures of DNA and its interactions with electrons, thus facilitating the investigation of DNA damage at the molecular level. In contrast, conventional methods such as fluorescence and electron microscopy to study specific molecular interactions and dynamics may not provide the same level of detail in terms of electronic states and charge transport mechanisms (Maeda, Matsumoto & Kawai, 2011; Elliott & Jones, 2018).

A novel approach to the study of PD is proposed, focusing on electron transport through DNA using microscopic research tools. Electron transfer is one of the fundamental keys to understanding the biological functions of damage repair mechanisms in DNA. In physics and chemistry, a number of experimental studies have generally focused on the electron transport properties of the materials under investigation. As STM is particularly well suited to the study of complex interactions between DNA and other biomolecules or nanoparticles that can affect electron transfer rates, the technique enables the investigation of molecular assemblies and the effects of external perturbations, such as electric fields or chemical modifications, on electron transport (Elliott & Jones, 2018; Fereiro et al., 2018). This capability is less accessible using conventional techniques, which may not provide the same level of insight into the dynamic interactions that occur in biological systems. The development of electrochemical STM further enhances the applicability of STM in biological research by enabling the study of electron transfer processes in situ under conditions that mimic physiological environments (Albrecht, 2012; Cucinotta, Rungger & Sanvito, 2012).

To our knowledge, despite numerous research using various microscopic techniques, including STM, to study electron transfer, there have been few studies using this particular technique to investigate the dynamics of electron transfer in abnormal or damaged DNA in PD. Our novel proposal for DNA research thus builds a bridge between physics and medical research. We propose to use STM as one of the microscopic tools, where STM measurement would provide us with information on electron transfer and authentic structural images at the nanometer scale in the DNA of PINK1 in PD. The microscopic properties of the DNA molecules measured by STM would provide invaluable information about the height profile, which indirectly corresponds to the tunneling current and provides topographical data with more precise measurements than the transmission electron microscope (Rafati & Gill, 2016).

By performing these experiments on normal and damaged DNA molecule samples in PD, we could investigate how electron transfer is affected by damage in DNA. The measurements of the height profile and the current–voltage (I–V) curve are essential for determining and explaining the differences in electron transfer between DNA molecules in PD. From the images of the DNA in PD measured with the STM, the brightness of the images on the DNA molecules can be used to determine how strongly the DNA molecules are electrically conductive. The difference in the brightness of the images will also help us to qualitatively recognize whether the differences in the DNA sequences can be identified by comparing the structures of the DNA. On the other hand, the I–V curves measured on the DNA molecules would give us information about the conductivity information of the molecules, i.e., whether the molecules have the properties of an insulator or electrical conductor. The differential conductance (dI/dV–V) curve patterns, particularly the peak positions in such curves, could serve as a discrete energy spectrum of DNA molecules and be characterized as a fingerprint (Shapir et al., 2010).

Conclusions

Our aim was to investigate previous studies using STM to study electron transfer in PD DNA through a systematic review of the literature. Most of the articles reviewed relate to biological, chemical and medical aspects of the investigations, in addition to epidemiological studies, drug treatment studies and more. Thus, many of the research studies focus on the biological and medical aspects of PD. Based on this systematic review, we propose to use STM as a new technique to study electron transfer in the DNA of PD from the perspective of physics.

STM imaging of DNA molecules and differential tunneling current detection of the DNA molecules between the atomically sharp STM tips, the base and a conducting surface could be used to establish an electronic fingerprint of different bases in many genetic diseases, especially PD. STM has a considerable advantage over other microscopy techniques in terms of resolution, as the surface topography can be imaged in much greater detail than with light microscopy. STM is also non-destructive compared to similar microscopic techniques, i.e., scanning electron microscopy and transmission electron microscopy, which destroy the samples under investigation due to their high penetration energy.

Supplemental Information

Supplemental Information 1 PRISMA checklist

Supplemental Information 2 List of publications for the screening

Supplemental Information 3 Comparison of experimental and review articles in previous studies, including study design, techniques, samples/models, and findings regarding electron transfer, DNA, and PD

For valuable discussions, the authors acknowledge Koichi Ichimura of Hokkaido University, Harison Rozak of Shibaura Institute of Technology, and Nor Safira Elaina Mohd Noor of the School of Electrical & Electronic Engineering, USM.

Additional Information and Declarations

Competing Interests

Author Contributions

Data Availability

Jafri Malin Abdullah is an Academic Editor for PeerJ.

Muhammad Hanif Che Lah conceived and designed the experiments, performed the experiments, analyzed the data, prepared figures and/or tables, authored or reviewed drafts of the article, and approved the final draft.

Mohammed Faruque Reza conceived and designed the experiments, authored or reviewed drafts of the article, and approved the final draft.

Shaharum Shamsuddin conceived and designed the experiments, authored or reviewed drafts of the article, and approved the final draft.

Isao Watanabe performed the experiments, analyzed the data, prepared figures and/or tables, authored or reviewed drafts of the article, and approved the final draft.

Jafri M. Abdullah conceived and designed the experiments, authored or reviewed drafts of the article, and approved the final draft.

The following information was supplied regarding data availability:

This is a systematic review/meta-analysis.

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
