# Peer review of "Systematic review of electron transfer study in DNA relevant to Parkinson’s disease and scanning tunneling microscopy"

_PeerJ, doi:10.7717/peerj.19807_

## Round 0.1 · original submission · Minor Revisions

Thanks for your patience with this review process, with this specific topic it was extremely hard to get experienced and willing reviewers.

Can you please repond to all the suggestions made by the reviewers in your resubmission.

Reviewer 1 ·

Basic reporting

The paper is well-written in clear, professional English, making it accessible to a broad scientific audience. The introduction is strong, providing a comprehensive overview of the topic and identifying relevant gaps in the literature. However, there are a few things that could be improved. For example,

1) the sentence in line 194 about “DNA being the optimal model for studying electron transport” might be supported by a reference.

2) the claim in line 348 that “very few studies use microscopic tools to investigate electron transfer in DNA” is misleading, as there are many studies utilizing such tools (including STM), although perhaps not in the context of Parkinson's disease. This sentence should be rephrased for accuracy.


Moreover, the graphical representations of data could be improved. For example,

1) the figures would be easier to navigate if the largest sections were consistently represented by the same color (for example, red), the second largest by another color (such as blue), and so on. Alternatively, if these sections followed each other in descending order (in any direction) it would also improve clarity. Applying at least one, or both strategies together would make the figures more representative.

2) It is obvious that the percentages were put manually, and it does not look good when small percentages are going to other sections. Also, white color is hard to see on some backgrounds. I would recommend authors use Python’s Plotly (https://plotly.com/python/pie-charts/) to improve their pie charts.

Experimental design

The paper outlines a well-structured search strategy, covering three databases. However, the rationale for selecting only these specific databases (ScienceDirect, EBSCOhost, and MEDLINE) is not fully explained, especially since the initial search resulted in a relatively small number of outcomes (13 and 8, respectively, for ScienceDirect and EBSCOhost MEDLINE). The authors should justify why these three databases were chosen and whether a broader search using more databases could have revealed additional relevant studies.

Validity of the findings

The review does a good job of identifying a key gap in the literature: the lack of STM usage in investigating DNA electron transfer in the context of Parkinson's disease. The main idea of this work is that adopting STM could offer new insights into the molecular mechanisms underlying Parkinson's disease, particularly in terms of DNA damage and repair mechanisms. While this is a valid and potentially impactful direction for future research, the paper would benefit from a more detailed discussion on how STM compares and why it offers advantages over other techniques.

Reviewer 2 ·

Basic reporting

In general, the manuscript is clearly structured with a good provision of context, with most citations appropriately referenced. However, there are some concerns regarding the style of writing, specifically on the ambiguous use of terminology and overly reductionistic statements, which might be a point of bias for the search strategy used in this systematic review. For example, it is unclear whether the authors are interested in studying the damage in nucleic DNA or in mitochondrial DNA - as both are inherited by the offspring, and both are implicated in PD pathogenesis. A comprehensive understanding of PD pathogenesis and the application of STM in biological samples would be beneficial in strategizing the literature search for this paper.

Experimental design

This systematic review provides an impactful research question and has identified a knowledge gap using the standard PRISMA guidelines. Nonetheless, a refined search strategy with good understanding of the existing pool of literature would benefit the study design.

Validity of the findings

No comment.

Reviewer 3 ·

Basic reporting

This is a well written review, although some of the sentences in the paper require rewriting to improve the grammar/clarity of the statements,
The authors need to explain what they mean about:
`The electron transport process is of fundamental importance for understanding the
biological processes of the damage repair mechanism in DNA.` There is no explanation offered. Is this electron transport within DNA or are they referring to the mitochondrial electron transport chain, I was very unclear after reading the paper.

Experimental design

The experimental design is appropriate.

Validity of the findings

The findings of the systematic review are valid and justified.

Additional comments

The introduction has some grammatical errors in the text and requires more explanation about the electron transport process and what precisely they mean by this and why it is important.

---

## Round 0.2 · accepted · Accept

All concerns of the reviewers were addressed and the manuscript is acceptable now.

Reviewer 1 ·

Basic reporting

The manuscript remains well-written in clear and professional English. All of my previous comments have been addressed, and I have no further suggestions.

Experimental design

With the authors' explanation of their choice of the three databases, the experimental design now appears solid.

Validity of the findings

Now the findings of the systematic review are well-supported.

Additional comments

I have no more comments and recommend the article for publication.